# Selection of Collagen Membranes for Bone Regeneration: A Literature Review

**DOI:** 10.3390/ma13030786

**Published:** 2020-02-09

**Authors:** Luca Sbricoli, Riccardo Guazzo, Marco Annunziata, Luca Gobbato, Eriberto Bressan, Livia Nastri

**Affiliations:** 1Department of Neurosciences, School of Dentistry, University of Padova, Via Giustiniani 2, 35100 Padova, Italy; riccardo.guazzo@gmail.com (R.G.); gobbatoluca@gmail.com (L.G.); eriberto@studiobressan.com (E.B.); 2Multidisciplinary Department of Medical Surgical and Dental Specialties, University of Campania “Luigi Vanvitelli”, 80100 Naples, Italy; marco.annunziata@unicampania.it (M.A.); livia.nastri@unicampania.it (L.N.)

**Keywords:** collagen membrane, guided bone regeneration, bone augmentation, biocompatible materials, dental implants

## Abstract

Several treatment modalities have been proposed to regenerate bone, including guided bone regeneration (GBR) where barrier membranes play an important role by isolating soft tissue and allowing bone to grow. Not all membranes biologically behave the same way, as they differ from their origin and structure, with reflections on their mechanical properties and on their clinical performance. Collagen membranes have been widely used in medicine and dentistry, because of their high biocompatibility and capability of promoting wound healing. Recently, collagen membranes have been applied in guided bone regeneration with comparable outcomes to non-resorbable membranes. Aim of this work is to provide a review on the main features, application, outcomes, and clinical employment of the different types of collagen membranes. Comparisons with non-resorbable membranes are clarified, characteristics of cross-linked collagen versus native collagen, use of different grafting materials and need for membrane fixation are explored in order to gain awareness of the indications and limits and to be able to choose the right membrane required by the clinical condition.

## 1. Introduction

Collagens are a family of different types of structural proteins found in many human tissues, such as skin, blood vessels, and bone. Collagen can be synthetized by many specialized cells in the human body, depending on the localization; fibroblasts are responsible for collagen production in the connective tissue while osteoblasts for the bone [1]. Collagen has many features, other than structural, including low immunogenicity, good hemostatic capacity, a chemotactic action on regenerative cells such as fibroblasts and osteoblasts and, lastly, good dimensional stability.

Collagen molecule alone is not stable. Therefore, in nature it is arranged into a triple helix structure, also called collagen fibril. Many fibrils are then arranged together with a covalent cross-linked bond to obtain a collagen fiber. There are different types of collagen, depending on the location and function. To date, more than 20 types have been classified. Collagen from type I to type IV are the most common in the human body and their localization is shown in Table 1 [1].

Collagen type I is also the major component of several commercially available collagen membranes (CM) developed as an evolution of the barrier membranes firstly introduced in the mid-80s [2] for regenerative purposes. The principle of the so called guided tissue regeneration was based on the ability of the membrane to exclude epithelial and connective cells in order to increase the ability of damaged periodontal tissue to regenerate, with new bone, periodontal ligament and cementum formation [3]. More recently the membrane barriers have been applied to regenerate bone for the development of the implant site (guided bone regeneration, GBR) [4]. These procedures are still widely employed in regenerative periodontal clinics [5] and to treat various intraoral bone defects, not limited to implant purposes [6,7].

The first successful use of resorbable membranes for GBR was reported in the early 1990s [8,9] and both natural and synthetic polymers have been used extensively with collagen being the mostly investigated [10]. These membranes are usually used in combination with autogenous or synthetic bone grafts, with or without screws and pins because they are incapable of maintaining defect space because of their lack of rigidity.

The use of (CM) was introduced to overcome the limits of non-resorbable membranes. The latter are technically more difficult to use because they cannot be left exposed to the oral environment and, if accidentally exposed, may lead to complications. Furthermore, non-resorbable membranes require a second surgery to be removed, with a consequent greater invasiveness for the patient [11,12,13,14].

Ideally, the biodegradation rate of the membranes should match the rate of new tissue formation with no residual materials left.

Currently, most resorbable membranes are made of collagen and there are a variety of membranes commercially available (Table 2). CM have been shown to stimulate the fibroblast DNA synthesis, and osteoblasts show high levels of adherence to CM surfaces. The in vivo biodegradation of CM takes place by endogenous collagenases into carbon dioxide and water [15].

One of the most interesting aspects of CM is that the speed of resorption may vary. It is important to be able to choose a membrane that maintains its structural integrity for the time necessary to the proliferation and maturation of the desired cells inside the wound. Commercially available CM provide different resorption time.

Apart from resorption time, membranes intended for bone regenerative purposes, should allow the achievement of several principles, published by Wang and Boyapati in 2006 [16] as “PASS”; primary wound closure without tension to enable proper healing by means of first intention and reduction of the risk of membrane exposure, angiogenesis to promote blood supply, space maintenance to create a bed for the undifferentiated mesenchymal cells and clot stability to allow for the proper development of these cells.

An “ideal” barrier membrane should present the following characteristics: biocompatibility (in order to prevent adverse reactions with the surrounding tissue and with the organism); tissue integration (to favor the embedding in the surrounding tissue and allowing a progressive integration of collagen fibers); dimensional stability (the positioning and shape of the membrane should remain unaltered till degradation); handling (the membrane should be managed and easily placed over the defect); selective permeability (the membrane should be able to exclude unwanted epithelial cells while allowing osteogenic cells to proliferate); space making function (in order to provide space for a stable blood clot, to allow bone regeneration) [17].

Researchers’ interest regarding non-resorbable membranes and collagen membranes varied during years. Nowadays, CM are the most employed and studied devices for bone regeneration for several indications, such as lateral bone augmentation, implant site development, ridge preservation and others [18,19,20]. Figure 1 shows the number of published articles by year for non-resorbable membranes, native collagen membranes, and cross-linked collagen membranes (Figure 1).

The aim of this paper is to compare different commercially available collagen barriers membranes and analyze their properties with the intention to clarify their clinical use.

## 2. Materials and Methods

For this narrative review, only studies published in English language were included, and the last search was carried out in December 2019. A literature search was performed in the PubMed/MEDLINE and ScienceDirect databases, without limiting the years of publication and including in vitro, in vivo and human studies and reviews that reported data on Collagen membranes for bone regeneration. Papers published before December 2019 and relevant to the topic were included. The following keywords were used in different combinations: “Guided Bone Regeneration,” “GBR,” “Ridge Augmentation,” “Barrier,” “Membrane,” “Polytetrafluoroethylene,” “PTFE,” “Collagen,” “Non-Resorbable,” “Resorbable,” “Cross-Linked,” “Native Collagen,” ‘‘Graft,”

‘‘Bone Grafts,” ‘‘Bone Substitutes,” “Autogenous Bone Grafts,” “Double Layer,” “Single Layer,” “Fixation,” “Pin,” Tacks,” “Screws.”

Titles and abstracts of the studies were screened independently by two reviewers (LN and LS) and categorized as suitable or not for inclusion. Full reports were reviewed independently for studies appearing to meet the topic of interest or for which there was insufficient information in the title and abstract to allow a clear decision. A hand search was also performed after checking references of the identified articles.

Included papers have been pooled in five different categories of clinical interest: Comparison between collagen membranes and non-resorbable membranes; native collagen versus cross-linked collagen membranes; use of grafts in conjunction with collagen membranes; use of a double layer of collagen membrane; need of fixation.

## 3. Results and Discussion

The electronic database and the hand search identified a total of 987 articles. Sixty-eight articles fulfilled the inclusion criteria. Some of the papers were included in more than one category of interest.

### 3.1. Collagen Membranes vs. Non-Resorbable Membranes

A wide range of membrane materials have been used in experimental and clinical studies to achieve guided bone regeneration (GBR) including polytetrafluoroethylene (PTFE), expanded PTFE (ePTFE), collagen, freeze-dried fascia lata, freeze-dried dura mater allografts, polyglactin 910, polylactic acid, polyglycolic acid, polyorthoester, polyurethane, polyhydroxybutyrate, calcium sulfate, micro titanium mesh [21], and titanium foils [12,21,22,23].

The first membranes made of polytetrafluoroethylene (e-PTFE; Teflon), had been shown to halt the migration of epithelial cells to the regenerating site where bone had to be produced [22,24,25,26].

With the presentation of the first successful GBR procedures and the subsequent wide and successful application of ePTFE membranes, this material became a standard for bone regeneration.

Current evidence supports the use of both non-resorbable membranes and resorbable membranes [19]. The main disadvantage of non-resorbable membranes was a higher rate of wound dehiscences [27,28], leading to a high occurrence of infections and adverse events in wound healing [29,30,31], for resorbable membranes, instead, limitations are a lack of space maintenance and a shorter degradation time. Because the longevity of the barriers’ function is an important aspect for the regenerative function, the loss of the structural integrity of a membrane because of the macrophage- and polymorpho-nuclear leukocyte-derived enzymatic activities may become a limit of bioresorbable devices [32,33].

Although the durability of the barrier effect may be diminished over the healing period, CM have several advantages such as a single-step surgical procedure, which decreases patient morbidity and the risk to the newly regenerated tissues, good tissue integration, with lower risk of membrane exposure, radiolucency that allows imaging of the regenerated bone during healing [34].

For bioresorbable and biodegradable membranes, additional criteria need to be fulfilled. Tissue reactions resulting from the resorption of the membrane should be minimal, these reactions should be reversible, and they should not negatively influence regeneration of the desired tissues [35].

Apart from the unnecessary second surgical intervention to remove the membrane, bioresorbable membranes offer some additional advantages: improved soft tissue healing, the incorporation of the membranes by the host tissues (depending on material properties), and a quick resorption in case of exposure, thus eliminating open microstructures prone to bacterial contamination and self-limiting infection [23].

In general, for their biological properties, soft tissue healing is improved in the presence of bioresorbable compared to non-resorbable membranes [27,36]. Recently, Turri et al. [37] have also shown that CM act as bioactive compartments rather than passive barriers, as they are involved in attracting cells into the wound area, which secrete signals for bone regeneration and remodeling, and they promote the expression of chemotactic factors, thus modulating the overall osteogenic process. Moreover CM may adsorb mediators and growth factors released from bone and cells, a molecular process that might enhance guided bone regeneration. Zitzmann et al. [27] compared the resorbable collagen membrane to the conventional expanded polytetrafluoroethylene material for guided bone regeneration in situations involving exposed implant surfaces. Over a 2-year period, 25 split-mouth patients were treated randomly with one of the two kinds of membranes. Changes in defect surface for both types of membranes were statistically significant (P < 0.0001); however, no statistical significance (P > 0.94) could be detected between the two membranes. The mean average percentage of bone fill was 92% for collagen membrane and 78% for e-PTFE membranes sites. In the latter group, 44% wound dehiscences and/or premature membrane removal occurred. Author concluded that CM were a useful alternative to the well-established expanded polytetrafluoroethylene membranes.

A disadvantage of CM is related to their unfavorable mechanical properties, which may lead to collapse into the bony defect; hence, their combination with a bone graft is recommended when clinically applied [38,39].

In a very recent systematic review [28] to test the evidence regarding the efficacy of lateral bone augmentation procedures, the most often used type of intervention for bone augmentation was the combination of a xenogeneic particulate grafting material with or without autogenous bone particles and a resorbable collagen membrane (CM). Meta-analyses using a native CM in conjunction with a xenogenic particulate grafting material as control treatment demonstrated that the defect height reduction was not significantly different compared to the combined data of the respective test groups (Figure 2). Significant differences, however, were observed in direct comparison with the second most common membrane, a non-resorbable ePTFE membrane in favor of CM for the primary outcome, vertical defect resolution [28]. These data were based on two included studies [13,40], and, since the use of an ePTFE membrane was considered to be the gold standard for GBR procedure at implant sites with dehiscence defects [18,28], have to be considered significant.

The amount of bone fill with the resorbable membrane was similar to that obtained with the ePTFE membranes for some authors [27], while other studies, in situations where no membrane exposures were noted, showed more favorable results in terms of bone formation using the ePTFE membranes compared to the bioresorbable ones [41].

### 3.2. Native vs. Cross-Linked Collagen

Natural CM are native materials, meaning that the natural collagen structure of the original tissue and thus their natural properties are preserved in a special production process. Naturally grown membranes exhibit especially good handling properties, such as pull and tear resistance, and a good adaptation to surface contours compared to membranes made of pressed collagen. A multi-stage cleaning process that removes all non-collagenic proteins and antigenic components, is used for the production. The resulting membranes exhibit a natural three-dimensional collagen structure of collagen type I and a lower proportion of collagen type III. This process includes several washing steps with different pH solutions to obtain neutralization and deantigenisation. At the end of the process a lyophilization and sterilization procedure is applied [42].

Natural membranes made of collagen have the major handicap of rapid in vivo degradation failing to provide the structural integrity necessary for the entire process of bone regeneration [42]. The benefits of a cross-linked collagen membrane results in a barrier of increased area and thickness, compared with the application of a single layer collagen membrane. Cross-linked CM can reduce bone graft resorption, as membrane degradation starts shortly after implantation. It has been suggested that a 1-month barrier function time for each millimeter of bone regeneration is needed [43]. Garcia et al. [44] in their study state that GBR procedures through resorbable CM achieve volumetric bone gains with no statistical significance between the cross-linked and the non-cross-linked membranes. Moses et al. [45] also reported no substantial difference in preserving hard and soft tissue volume between cross-linked and non-cross-linked membranes. Nevertheless, they performed significantly better if non-exposed.

In terms of biocompatibility, tissue integration and postoperative complications the results of Garcia’s review suggest that non-cross-linked membranes present better results (Figure 3). The porous structure of non-cross-linked CM is suitable for the formation of transmembrane blood vessels, which may also facilitate membrane resorption [42].

Calciolari et al. [17] showed as native CM, derived from porcine type I and III collagen, were biocompatible and inert, did not elicit an inflammatory or foreign body reaction, and were able to promote the bone regeneration process. Membrane integrity was well maintained during the first 14 days but, at 30 days, pronounced signs of degradation, high levels of remodeling and a significant reduction in thickness were identified. Similar findings were published by Moses et al. [45] showing a significant reduction in membrane thickness from 14 to 30 days of healing, as well as a significant reduction in the total amount of collagen. Nevertheless, at 30 days, bone formation markers (alkaline phosphatase, bone sialoprotein, osteopontin), a mesenchymal cell marker (vimentin) together with histological features suggested that bone formation was occurring incorporating fragments of the degraded collagen fibers [45,46].

To improve the resistance to degradation and prolonging the effect of resorbable CM, physical, chemical and enzymatic processes were developed to improve durability by cross-linking the existing collagen fibers and thus creating resorbable cross-linked CM [47,48,49].

The formation of collagen cross-links is due to the presence of two aldehyde-containing amino acids which react with other amino acids in collagen to generate difunctional, trifunctional, and tetrafunctional cross-links. The collagen molecules assembled in the naturally occurring fibrous polymer is a prerequisite for the development of these cross-links. When this is achieved, cross-linking occurs in a spontaneous, progressive fashion. The chemical structures of the cross-links dictate that very precise intermolecular alignments must occur in the collagen polymer. This seems to be a function of each specific collagen because the relative abundance of the different cross-links varies markedly, depending on the collagen tissue origin [50].

Various chemical and physical cross-linking methods, such as ultraviolet light, glutaraldehyde (which is a reference agent for the cross-linking reactions), glutaraldehyde plus irradiation, hexa-methylenediisocyanate (HMDIC), diphenylphosphorylazide, and enzymatic ribose cross-linking, have been used to boost the biomechanical properties of the collagen fibers [15]. The manufacturing process involves the extraction of collagen into monomeric fibrils, which are then reconstructed and cross-linked to form an improved collagen-based biomaterial [51].

The in vivo degradation of collagen biomaterials can be controlled by this cross-linking reaction. Glutaraldehyde (GA) reacts with the amino groups of the side-chains of collagen molecules, creating a framework in the material that improves the mechanical and biological stability. Some problems related to GA cross-linking, such as polymerization of GA monomers in solution leading to heterogeneous cross-linking and cytotoxicity, have been overcome by continuous reaction with GA at low concentrations. This method may produce a material with the same pattern of degradability using smaller amounts of GA, thus avoiding cytotoxic effects. Progressive treatment with low concentrations of GA is believed to induce more homogeneous reactions in the collagen matrix [52].

The degree of cross-linking of collagen fibers affects the rate of degradation with more cross-linking leading to slower degradation and vice versa [48,53].

Rothamel et al. [54] compared the biodegradation of differently cross-linked CM to a native collagen membrane in rats. They observed at 2, 4, 8, 16, and 24 weeks the membrane behavior with regard to vascularization, tissue integration, inflammatory response during resorption. Highest vascularization and tissue integration were noted for native collagen followed by cross-linked membranes, some of which exhibited a foreign body reaction during resorption.

Moses et al. evaluated the biodegradation of three different commercially available CM. Statistically significant differences in the amount of residual membrane material were recorded within each membrane (among different time moments) and among different membranes at the same time moments. At 28 days, the least amount of residual collagen area was observed in the non-cross-linked membranes (13.9% ± 10.25%), followed by the glutaraldehyde cross-linked (24.7% ± 35.11%) and ribose cross-linked (91.3% ± 10.35%) groups. Residual membrane thickness, expressed as the percentage of baseline thickness, presented a similar pattern [45].

Cross-linked CM can reduce bone graft resorption, as membrane degradation starts shortly after implantation. It has been suggested that a 1-month barrier function time for each millimeter of bone regeneration is needed [43].

Chemically cross-linked CM have longer degradation times but also have significantly higher membrane exposure rates, up to 70.5% [45,55,56].

Wound dehiscence with membrane exposure has a substantial negative effect on GBR. Tal et al. [55] also demonstrated that both cross-linked and non-cross-linked membranes were resistant to tissue degradation and maintained continuity to ensure bone regeneration, however, none of the membranes was resistant to degradation when exposed to the oral environment with a substantial loss of regenerative effect.

In a recent human study, comparisons of cross-linked to non-crosslinked conventional collagen membrane, placed at implant dehiscence sites showed that both membranes yielded comparable bone regeneration results. Nevertheless, it was concluded, that premature membrane exposure of the cross-linked membrane might impair soft tissue healing, or may even cause wound infections [57].

### 3.3. Collagen Membranes in Conjunction with Graft

As previously stated, resorbable membranes have some limits in their mechanical properties, which may lead to a low space maintenance [38,39]. CM used alone, without particulate matter or graft blocks for GBR, usually result in membrane compression into the defect space by overlying soft tissues [24].

For this reason, CM are also frequently combined with block bone grafts [58,59,60,61] with or without autogenous bone chips for graft consolidation, or xenografts or alloplastic bone substitutes [62,63].

Autogenous bone grafts provide some favorable characteristics for bone regeneration [64], for example osteoconductivity [4], osteogenicity [65], and osteoinductivity [66], but their harvesting may increase patient morbidity due to the additional surgical procedures [67] and it is reported a fast degradation, with the loss of the space making function.

In order to overcome this disadvantage and to obtain a slower degradation rate, bone substitute materials have been extensively evaluated [68].

Benic et al. [69] showed that a block bone substitute in combination with a collagen membrane and fixation pins was superior to a particulate bone substitute with a collagen membrane and fixation pins with respect to the thickness and to the vertical gain of the augmented hard tissue after 6 months of healing in comparison to the sites grafted with a particulate deproteinized bovine bone mineral (DBBM).

While bone blocks are more indicated to obtain vertical bone increases, resorbable membranes in combination with particulated bovine bone can be used for the augmentation of horizontally deficient ridges [70]. Bone substitutes can be mixed with particulated autogenous bone to add more osteogenic factors [71,72].

Meloni et al. [20] showed that patients, having less than 4 mm of residual horizontal bone width were selected and consecutively treated with resorbable CM and a 1:1 mixture of particulated anorganic bovine bone and autogenous bone, 7 months before implant placement. An average horizontal bone gain of 5.03 ± 2.15 mm (95% CI: 4.13–5.92 mm) was obtained.

CM and different grafting materials can be adopted also to perform ridge preservation techniques. Although a reduction of alveolar ridge resorption can be achieved with the application of resorbable CM without bone grafts, Iasella et al. [73] showed that extraction sites with bioresorbable membrane showed less vertical and horizontal bone loss and greater bone fill.

A recent preclinical study assessed the in vivo performance of a collagen-containing equine block, of a deproteinized bovine bone mineral (DBBM) block and of particulate DBBM used for GBR with simultaneous implant placement [74]. GBR with bone substitute blocks lead to higher ridge dimensions than empty controls. The equine block with collagen membrane obtained the most favorable outcomes in hard and soft tissue contours followed by deproteinized bovine bone mineral block and particulated deproteinized bovine bone mineral with collagen membrane.

### 3.4. Single Layer vs. Double Layer

CM can be used as a single layer or with a double layer, by overlapping two membranes. Buser et al. [75,76] for the first time proposed the use of a double layer for guided bone regeneration. The rationale for the use of a double membrane layer is the reduction of micro-movements and the best stabilization of the graft, optimizing the sheltering in the area to be regenerated. Kim et al. [77] verified the improvement in bone block stability for hard tissue regeneration using the single or double layer on a rabbit model. They reported a statistically significant higher bone volume after 4 months between the double layer and the single layer groups, but no statistically significant difference at 6 months between the two groups. In 2017, Choi et al. [78] performed a human study to test the alveolar ridge preservation using single- or double-layer collagen membrane. The two groups showed no difference in terms of preservation of horizontal and vertical dimensions of the alveolar socket.

Kozlovski et al. [46] investigated whether there was a difference between a single and a double membrane layer in the repair of bone defects in a rat model. The study showed a statistically significant difference in the collagen area (0.09 ± 0.05 mm^2^ vs. 0.047 ± 0.034 mm^2^ respectively in the double and single layer.) On the other side, no statistically significant difference was seen in the reduction of thickness between 4 and 9 weeks.

Some authors have reported better results of bone formation when using a native collagen membrane in a double-layer technique instead of using it in a single layer in GBR of horizontal ridges [76]. Von Arx and Buser [76] in 2006 conducted a study similar to Antoun et al. [79] by covering horizontally grafted sites by bone blocks and bovine bone particles and covered by a double membrane native collagen membrane. This technique allowed, according to the authors, a better protection of the graft and increased the stability of the membrane. Moreover, this double collagen barrier provided an increase in the survival of the membrane and a prolonged protection of the horizontally grafted ridge by a bone block. Results showed a reduction of only 7% of the total width of the graft after 6 months.

Other studies attempted to analyze the effects of single- and double-layered CM on the bone resorption and augmentation efficacy of onlay block bone grafts. The results indicated that the double-layer technique was associated with a decreased collapse tendency and a higher bone density of onlay bone graft than the single-layer membrane technique, explained by the protective effects of graft resorption during the healing time [75,76,77].

Double-layered membranes are thought to retain their barrier function for a longer period of time. However, in a different study, designed to evaluate histologically and histomorphometrically the bone regeneration in critical size calvarial defects in rats grafted with either a deproteinized bovine bone mineral alone or in combination with a single or double layer of native bilayer collagen membrane, a single-layer collagen membrane was sufficient to allow the exclusion of connective tissue cells or cells from muscular origin, and adding a second membrane layer did not demonstrate a further increase in the amount of new bone formation compared to a single-layer collagen membrane. The author commented that although a lack of significant improvement in efficacy, this double cover had an undoubtful advantage in the stabilization of the graft [80].

### 3.5. Fixation vs. Non-Fixation

Micromotion of the membrane or of the contained graft can influence the volume of the augmented site during the healing period, especially with particulate bone grafts.

Stability can be achieved by fixing the membranes to the local bone using miniature pins of polylactic acid, osteosynthesis screws, and titanium pins or using ligatures tying the membrane to the soft tissues adjacent to the site of regeneration.

Urban et al. [81] compared different treatment groups of GBR, including procedures with membrane fixation or not. The study showed that any form of stabilization of one-wall horizontal bone augmentation resulted in a better stability of the graft. When the most commonly used GBR method (particulate bone grafting combined with a resorbable collagen membrane with no further membrane stabilization) was performed, the collagen membrane was able to prevent the ingrowth of soft tissue into the graft material, however, it lacked in form stability. Groups with membrane stabilization always showed better outcomes, preventing graft migration and membrane collapse.

Despite the use of pins for membrane stabilization, considerable displacement of the particulate grafting material occurs both during flap suturing and during the subsequent healing period [13,69].

Mir-Mari et al. [32] showed that wound closure induced displacement of the bone substitute resulting in a partial collapse of the collagen membrane in the coronal portion of the augmented site. The use of fixation pins in combination with particulated bone substitute and collagen membrane and the application of the block bone substitute with collagen membrane significantly better performed in the dimensional stability of the augmented site, as compared to GBR trough particulated bone substitute and collagen membrane. The investigators found that wound closure induced a considerable displacement of particulate deproteinized bovine bone mineral DBBM resulting in a partial collapse of the collagen membrane. The additional use of fixation pins or the use of block instead of particulated grafting material permitted reducing the amount of membrane collapse. Stability of the augmented site could be enhanced either by stabilizing the barrier membrane or by providing adequate support to the membrane with a bone block, even if the sites augmented with DBBM block and collagen membrane without pins exhibited an average loss of thickness of 20% caused by the displacement of the block graft during wound closure. The sites augmented with DBBM block + collagen membrane + fixation pins were able to maintain the space during flap suturing [32,82].

In a previous clinical study, GBR procedures with resorbable or non-resorbable membranes were performed with or without the use of polylactide pins [13]. When membrane fixation was provided, a significantly higher success of GBR was found in terms of frequency of postoperative complications and reduction in the size of the peri-implant defects, as compared to GBR without membrane fixation [82].

Some author stated that the lack of titanium reinforcement for the collagen membrane can be overcome by an accurate fixation of the membrane with titanium pins on both the lingual/palatal and the vestibular side. With a secure fixation the membrane immobilizes the graft material until the complete resorption, allowing the formation of the desired amount of bone [20].

## 4. Summary and Conclusions

Guided bone regeneration represents a well-established procedure for bone augmentation, and it can be carried out using several kinds of membranes and grafts. Membranes should be chosen for each clinical case according to the desired biodegradation characteristics. Resorbable and non-resorbable membranes differ in terms of clinical and technical handling, rate of complications, and expected long-term outcomes. Among the different available resorbable membranes, collagen membranes may account on a very sound scientific background and a largely validated clinical employment. They offer benefits in terms of no need to remove the membrane at second-stage surgery, favorable biologic attributes and similar long-term performance at dehiscence-type defects. CM show to better interact with soft tissues, allowing more oxygen exchange, micronutrient pass and blood perfusion, and favoring cell proliferation and differentiation. In the collagen membrane family, the choice between cross-linked and non-cross-linked CM may affect their clinical use. The degree of cross-linking of collagen fibers, indeed, have shown to affect the degradation rate, and the preservation of the underlying bone graft. Longer degradation times and membrane resistance to resorption, however, are also linked to a significantly higher exposure rates for cross-linked collagen membranes, and, sometimes, a foreign body reaction during resorption.

Grafts have been successfully applied in combination with membranes in several human bone regeneration and augmentation studies. The use of deproteinized bovine bone matrix, in particular, does not enhance per se the ability of the membrane to promote bone formation, but it demonstrates osteoconductive properties and provides mechanical support to the barrier preventing its collapse into the defect. In clinical practice, indeed, successful regeneration is obtained only when cell occlusion and space-maintaining exist for the healing time needed to osteoprogenitor cells to repopulate the defect. Probably, in case of vertical or non-supporting/non-containing defects, collagen membranes should also be supported by a block graft material to enhance the maintaining and the stabilization of the regenerative space. Membrane fixation, moreover, appears to favor more predictable outcomes, preventing the dislocation of both membrane and graft during the suturing and the post-operative phases.

In addition to the abovementioned membrane features and technical aspects, some other principles have to be remembered for a successful GBR treatment such as a well-designed flap, tension-releasing procedures, a complete wound closure without any tension or compression during healing. An uneventful healing process is always desirable in clinical practice; however, it is not uncommon observing membrane exposure, especially during the early healing stages. As the membrane becomes exposed to the oral environment, a faster degradation occurs, with a high risk of bacterial membrane colonization that may significantly reduce the final regeneration outcome.

In conclusion, collagen membranes show advantageous biological and clinical features compared to both non-resorbable and other resorbable membranes, but they are not free from possible complications. Only the deep knowledge of the features of such biomaterials and the relative surgical procedures may allow clinicians to perform the right choice, in order to maximize the success rate of their clinical procedures.

## Figures and Tables

**Figure 1 materials-13-00786-f001:**
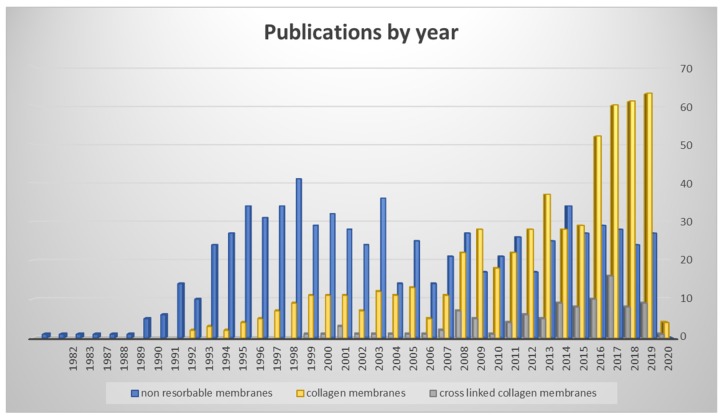
Publications by year on non-resorbable membranes, collagen membranes, and cross-linked membranes (data from PubMed Library). The chart clearly shows the increase of publications toward collagen membranes.

**Figure 2 materials-13-00786-f002:**
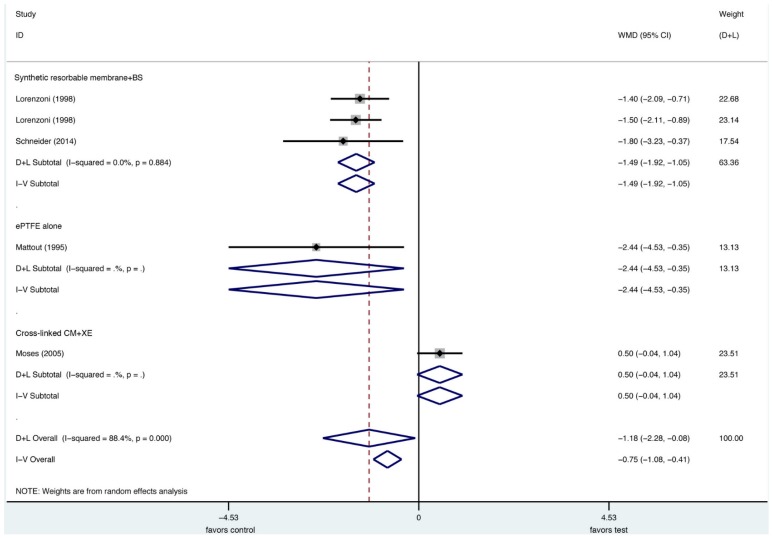
Forest plot illustrating the results in terms of defect height reduction from the meta-analysis of all trials with a non-resorbable expanded polytetrafluorethylene membrane in combination with any bone substitute (ePTFE + BS) as a control. CM: collagen membrane. (From Thoma, DS, Bienz, SP, Figuero, E, Jung, RE, Sanz-Martín, I. Efficacy of lateral bone augmentation performed simultaneously with dental implant placement: A systematic review and meta-analysis. *J Clin Periodontol*. 2019; 46(Suppl. 21): 257–276. With permission).

**Figure 3 materials-13-00786-f003:**
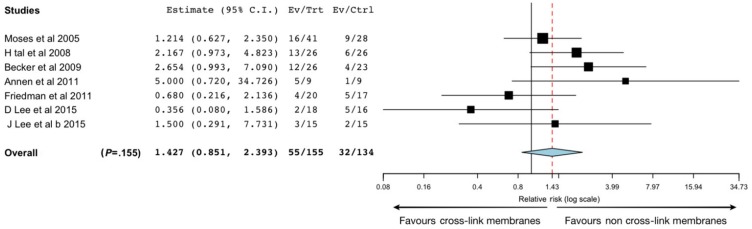
Analysis and forest plot for the results of the included studies that determined membrane exposure. Ev/Trt represents the test (cross-linked membranes) group, while Ev/Ctrl represents the control (non-cross-linked membranes) group. Red line represents the average for all results, and the vertical black line represents the no-effect line (from Garcia J, et al. Effect of cross-linked vs non-cross-linked collagen membranes on bone: A systematic review. *J Periodontal Res*. 2017;52(6):955–964. With permission).

**Table 1 materials-13-00786-t001:** Main types of collagen.

Collagen type I	90% of total collagen, mostly found in all main connective tissue such as skin, tendons, ligaments, bone, cornea and periodontal connective tissue
Collagen type II	Found in cartilage, intervertebral discs and vitreous body
Collagen type III	Mostly found in cardiovascular system and granulation tissue
Collagen type IV	In the basal membrane

**Table 2 materials-13-00786-t002:** Non-exhaustive list of available collagen membranes for clinical use (n.d. = not declared).

Commercial Name	Produced By/For	Origin	Cross-Link	Barrier Effect (Weeks)
Biomend	Collagen Matrix Inc.	Bovine Tendon	Yes	8
Biomend Extend	Collagen Matrix Inc.	Bovine Tendon	Yes	18
Copios Extend	Collagen Matrix Inc.	Porcine Dermis	No	24–36
Osseoguard	Collagen Matrix Inc.	Bovine Tendon	Yes	26–38
Bio Gide	Geistlich Pharma Ag	Porcine Dermis	No	24
Mem-Lok RCM	Collagen Matrix Inc.	Bovine Tendon	Yes	26–38
Mem-Lok Pliable	Collagen Matrix Inc.	Porcine Peritoneum	Yes	12–16
Ossix Plus	Datum Dental Ltd.	Porcine Tendon	Yes	16–24
Creos Xenoprotect	Nobel Biocare	Porcine	No	12–16
Biocollagen	Bioteck S.P.A.	Equine Tendon Type I Collagen	No	4–6
Heart	Bioteck S.P.A.	Equine Pericardium	No	12–16
Cytoplast	Collagen Matrix Inc.	Bovine Tendon Type I	Yes	26–38
Collatape	Zimmer -Biomet	Bovine Collagen	No	1–2
Jason	MBP Gmbh -Botiss Biomaterials	Porcine Pericardium	No	8–12
Collprotect	Botiss Biomaterials	Porcine Dermis	Yes	4–8
Dynamatrix	Keystone Dental	Porcine Submucosa	No	n.d.
Ez Cure	Biomatlante	Purified Porcine-Based Type I And III Collagen	Yes	12
Conform	Ace Surgical Supply Company	Bovine Type I Collagen	Yes	12–16

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
