# Peer review of "Selection of Collagen Membranes for Bone Regeneration: A Literature Review"

_materials, 2020, doi:10.3390/ma13030786_

Round 1
Reviewer 1 Report
This manuscript untitled “Selection of Collagen Membranes for Bone Regeneration: A Literature Review” was a makes a narrative review of collagen membranes. Aim of this paper is quite interesting. Generally, there are grammatical errors in this manuscript. It is recommended that it would be revised again by English scientific writer.
Additions are recommended:
What is the novelty of this paper? Please clarify in the appropriate section.
Page 4 line 142 - this sentence lacks the final score.
Page 4 line 158 – Please standardized the “P”, because in the same sentence the “P” was presented in two ways “P < .0001” Without zero and with “P> 0.94”.
Page 5 line 203 “non-cross-linked” remove the spaces for “non-cross-linked”
Page 7 line 276 – You have double “or”, remove one or.
The references are not well inserted in the manuscript, and standardized:
Page 2 line 52, 272 - [8],[9] must change for [8, 9] Page 5 line 196 and 198. Garcia et al. [42]….. and Moses et al. [44]. In these sentences the referenced authors lack et al. as well as the reference number. It should be noted that when identifying the study, the study should come immediately after et al. Page 6 Line 210, line 243, 248, 261, 310, 313, 316, 320, 363 and more. Moses et al. This fact should be standardized throughout the text, as in some cases it is with the author, others are at the end of the sentence. et al. must be italicized et al.
Table 2. you need standardized this table, because the some words are uppercase and others lowercase, “no”,”No”, “porcine”, “Porcine”, “equine”, Equine”” bovine”, Bovine”, “yes”, “Yes”…
Author Response
Kind Reviewer,
We really appreciated all your suggestions to improve our manuscript.
Our intention when we started to plan and write our manuscript “Selection of Collagen Membranes for Bone Regeneration: A Literature review” was to produce a narrative review with some clinical questions kept in mind, in order to help the clinician during the choice of the membrane and technique to regenerate bone. This link between scientific background and the potential clinical aspects, to the best of our knowledge, weas not present in a single article.
There were several typing mistakes. We apologize.
In details, we corrected all the mistakes you underlined (pag 4 line 142 and line 158, pag. 5 line 203, pag 7 line 276). We standardized the style of the references when an author was cited in the sentence. We add the missing et al.
We adjusted the format of the Table n° 2.
Hoping that the revision we wrote could be acceptable for the journal “Materials”,
our best regards
Livia Nastri
Luca Sbricoli
Riccardo Guazzo
Marco Annunziata
Luca Gobbato
Eriberto Bressan
Reviewer 2 Report
The manuscript “Selection of Collagen Membranes for Bone Regeneration: A Literature Review” aimed to evaluate collagen membrane according to their most appropriate clinical use.
This manuscript is a general review of things that are already known and provides no new information or conclusion that could be useful for the reader in the use of barrier membranes for GBR.
Several major issued needs to be addressed
In the entire manuscript the types of collagen are named with either roman or Arabic numerals: for example, in Table I – Arabic numerals are used, but in the row below (43), roman numerals are employed („Collagen type I”). In Table II, Arabic and roman numerals are used for naming collagen types. In the literature, the roman numerals are mostly used for differentiating the types of collagen. The authors should decide for either roman or Arabic numerals.
In the introduction, in rows 89 to 103, well-known indications for postoperative care are provided, without any specificity and no interest to the reader. This part should be replaced with a more comprehensive description on the structure, the biocompatibility or the degradation of collagen membranes or with de requirements for an „ideal” barrier membrane for GBR.
The aim („Aim of this paper is to review features and clinical suggestions to choose the right collagen membrane and the most appropriate use.”) is unclear and need to be rephrased. The author’s objective was probably to compare different commercially available collagen barriers membranes and analyze their properties with the intention to clarify their clinical use.
In Material and Methods key words or mesh terms used for the search are not provided. A more detailed search strategy is more than welcomed.
The Results paragraph should be beater structured. Also the number of articles found/assessed is not mentioned.
However, the manuscript lack of novelty and its organization is poor. The subheadings from the results paragraph are belonging to the Discussion paragraph.
Somme other important issues such as the enhancement of mechanical properties and bioactivities of the collagen membranes need to be also addressed.
Reviewer 3 Report
The authors have reviewed the use of different forms of collagen for guided bone regeneration. This area of research has been well documented and there is lot of literature available. This review is a small contribution to the literature.
The manuscript is well written and organized but needs improvements.
Please include images and graphs to give better over view.
Charts can include how many pubs vs years. Research trend towards non resorbable membranes vs native collagen vs crosslinked collagen etc.
Images can include reprints (with permission) from few interesting studies reviewed such as- bone formation comparison, effect of crosslinking on degradation rate etc
Include price comparison between different membrane materials (ePTFR, CM , crosslinked CM
Other corrections-Please use acronyms first time they appear. CM for collagen membranes was mentioned in line 168, they first appear in line 57.
GBR first appears in line 49 please use acronym here first
Reference missing for CM application lines- 79-87
Round 2
Reviewer 2 Report
The new version of the manuscript showed significant improvements and the authors addressed the prior mentioned shortcomings.
Reviewer 3 Report
The authors have addressed my comments and have improved quality of the manuscriipt